# The Spatial Diffusion of Cherry Leaf Roll Virus Revealed by a Bayesian Phylodynamic Analysis

**DOI:** 10.3390/v14102179

**Published:** 2022-10-01

**Authors:** Jianguo Shen, Jing Guo, Xihong Chen, Wei Cai, Zhenguo Du, Yongjiang Zhang

**Affiliations:** 1Fujian Key Laboratory for Technology Research of Inspection and Quarantine, Technology Center of Fuzhou Customs District, Fuzhou 350001, China; 2Institute of Plant Virology, Fujian Agriculture and Forestry University, Fuzhou 350002, China; 3Chinese Academy of Inspection and Quarantine, Beijing 100121, China

**Keywords:** cherry leaf roll virus, Bayesian phylodynamics, spatial diffusion, Markov jump

## Abstract

Cherry leaf roll virus (CLRV) is an important plant pathogen that causes severe and detrimental effects on cherry and other fruit plants. Despite recent progress in plant pathology, molecular biology, and population genetics of CLRV, the spatiotemporal spread of this virus remains poorly studied. In this study, we employed a Bayesian phylodynamics framework to investigate the spatial diffusion patterns of CLRV by analyzing the coat protein gene sequences of 81 viral isolates collected from five different countries. Consistent with the trade of cherry, our Bayesian phylodynamic analyses pointed to viral origins in New Zealand and identified multiple migration pathways between Germany and other countries, suggesting that Germany has played an important role in CLRV transmission. The results of our study will be useful in developing sustainable management strategies to control this pathogen.

## 1. Introduction

Cherry leaf roll virus (CLRV) is a subgroup C nepoviruses belonging to the subfamily *Comovirinae* of the family *Secoviridae* [1]. CLRV has a very wide host range comprising diverse woody and herbaceous plants from at least 28 distinct genera [2]. Since it was first noted in the United States, severe economic losses caused by CLRV have been reported for cherry in England and Germany, red raspberry in New Zealand, and walnut in Hungary [3,4,5,6,7]. Particularly, birch leaf-roll disease caused by CLRV has emerged as a major constraint for forest health in Fennoscandia and Germany, causing significant economic losses [2,8].

The genome of CLRV consists of two single-stranded positive-sense RNAs encapsulated separately in isometric particles around 28 nm in diameter. The two RNAs, called RNA1 and RNA2, each encode a single polyprotein. The polyprotein encoded by RNA1 is cleaved into a proteinase-cofactor (PCo), a nucleotide-binding helicase (Hel) containing a nucleoside triphosphate-binding protein-domain (NTB), genome-linked protein (VPg), proteinase (Pro), and the viral replicase (Pol), whereas the polyprotein specified by RNA2 is processed into the coat protein (CP) and the movement protein (MP) [9]. RNA1 and 2 have an almost identical 3′-noncoding region (NCR) that is unusually long, with more than 1500 nucleotides.

Although only a few CLRV isolates have been sequenced completely [3,9], the variation of this virus has been examined using serological or by biological methods [3]. In addition, phylogenetic reconstructions have been attempted using partial sequences of CLRV [2,3,10]. Overall, these studies revealed that CLRV has six major phylogenetic clusters [10]. Interestingly, CLRV isolates from the same host species, but not those from the same geographical origins, tend to form the same cluster. This is explained by the fact that CLRV is naturally transmitted by seed, pollen and mechanical inoculation, which imposes a barrier to inter-species transmission of this virus [4,10]. However, CLRV isolates from different host species may cluster together and those from the same host species may be grouped into distinct clusters, implying that inter-species transmission can occur in some cases [10]. The most recent example supporting the later idea comes from CLRV infecting birch. In both Germany and Fennoscandia, birch CLRV shows an unexpectedly high genetic diversity. Isolates recovered from a very limited region can be grouped into more than one cluster [2,8].

The evolutionary dynamics of CLRV are poorly understood. However, Bayesian phylogenetic reconstruction methods were recently applied to tomato ringspot virus (ToRSV), a subgroup C nepovirus closely related to CLRV. This study estimated that the substitution rate of ToRSV is 2.74 × 10^−3^ subs/site/year (95% credibility interval 1.23 × 10^−4^–6.23 × 10^−3^) [11]. This rate is comparable to those of animal-infecting RNA viruses, suggesting that nepoviruses have rapid evolutionary dynamics.

The spatial dynamics of CLRV, especially its transmission and spread, have not yet been investigated, although such information is vital in developing sustainable management schemes. In this study, we aim to solve these problems with Bayesian phylogenetic reconstruction methods.

## 2. Materials and Methods

The CP gene sequences of CLRV isolates were collected from the GenBank. Three sequences with only isolate from the country were removed to result in a final data set consisting of 81 CP sequences (Appendix A). The isolates came from five countries: Finland (*n =* 35), France (*n =* 10), Germany (*n* = 18), New Zealand (*n =* 15), and the United States (*n =* 3). Multiple sequence alignment was performed using the codon-based MUSCLE algorithm [12], implemented in MEGA 11 [13]. No significant signals of recombination were identified by any four of the seven algorithms in the RDP package [14], with an associated *p*-value of 10^−6^ and thus the complete data set was used for all our analyses below.

To infer the evolutionary rate and timescale of CLRV, we performed a Bayesian phylogenetic analysis in BEAST 1.8.4 [15] with the SYM+G4 substitution models, which was determined by ModelFinder [16] with Bayesian information criterion. Using marginal likelihoods estimated by path sampling [17], an uncorrelated lognormal relaxed molecular clock model and constant coalescent tree prior was shown to fit our data well (Appendix A). To assess the temporal signal in the data set, we first regressed root-to-tip distances against date of sampling using TempEst 1.5 [18] and then confirmed the presence of the temporal signal using a clustered date-randomization test [19]. However, we found no evidence of temporal signal in our data set and therefore a uniform prior of 1.23 × 10^−4^–6.23 × 10^−3^ substitutions/site/year was applied for the absolute substitution rate of the CP gene of CLRV, based on a previous estimate for ToRSV, as described above [11]. Three Markov chain Monte Carlo runs were performed for 100,000,000 generations (sampling every 10,000 states). Convergence and mixing were assessed using Tracer 1.71 [20], with a burn-in period of 10% of the total length. To verify the reliability of the inferred location at the root node, we produced 10 replicate data sets in which the location states were randomly swapped among the sequences and compared the results with that from the original data set.

To investigate dispersal pattern of CLRV across the different geographic regions, we modeled geographic regions as discrete trait using a discrete phylogeographic diffusion model in BEAST. We used the same substitution models as described above. Posterior distributions of parameters were estimated by Markov chain Monte Carlo sampling, as described above. Four countries, Finland, France, Germany, and New Zealand were selected and coded as discrete states. The United States CLRV population was excluded from analyses due to an inadequate sample size. A Bayesian stochastic search variable selection procedure (BSSVS) [21] was employed to allow for transitions between these countries. In addition, the complete Markov jump history was also recorded independent of the BSSVS runs [22]. The best-supported pairwise diffusions were assessed using Bayes factors (BF) in SPREAD3 0.97 [23]. Rates were considered statistically supported when BF > 3.0 and the indicator value of the corresponding > 0.50. The BF also provides the strength of the evidence as follows: support, 3 < BF < 20; strong support, 20 < BF < 150; very strong support, 150 < BF < 1000; and decisive support, BF > 1000 [24]. To avoid the influence of potential sampling biases on migration inference, we repeated Bayesian analyses using a bootstrapping approach to standardize sample sizes. For each bootstrap replication, we randomly selected 10 sequences (the actual size of the France population) without replacement from each country. Each of these datasets was analyzed using BEAST, as described above.

To further investigate the migration history of CLRV, we analyzed the inferred load and direction of migration over time using a reference python script by Brynildsrud et al. [25] (https://github.com/admiralenola/globall4scripts, accessed on 7 September 2022). For this analysis, migration events were assumed to occur on nodes. Although such an assumption might result in a slight bias towards inflated ages of migration events, the bias is negligible for later migrations [25].

## 3. Results and Discussion

Our time-scaled maximum-clade-credibility tree showed that CLRV isolates could be grouped into four lineages with high posterior probabilities (>0.95, Figure 1). In each lineage, the majority of isolates share the same original host. This coincides well with the proposed phylogenetic groups A-E by Rebenstorf et al. [10]. CLRV was first described (under the name of elm mosaic virus) in the United States [6,7] and was not officially named until 1955 [4,26]. Our Bayesian phylogenetic analysis placed the most recent common ancestor of CLRV lineages in 1920 CE (95% credibility interval 1773–1991, Figure 1), slightly earlier than the documented emergence of this virus. Our estimate of the substitution rate of the CLRV was 4.05 × 10^−3^ subs/site/year (95% credibility interval 1.19 × 10^−3^–6.23 × 10^−3^), which is considerably higher than those seen in other plant viruses [27]. However, these are not real estimates because the data set lacked temporal structure. Further investigations should be based on time-structured data of this virus, which will result in more reliable estimates of substitution rate from Bayesian analysis.

Our Bayesian phylogeographic analysis places the root of the tree in New Zealand, with a posterior probability of 0.77 (Figure 1). This is outside the range of probabilities (0.04–0.58) obtained in the analyses of the randomized results (Appendix A), providing further support for the reliability of New Zealand at the root node.

Four migration pathways were identified during the spatial diffusion of CLRV (Figure 2a), with mean rates between 0.82 and 1.96 migration events per lineage per year (Appendix A). The highest rates of viral migration were observed from Finland to Germany, whereas the lowest migration rate was observed from New Zealand to Germany (Appendix A, Figure 2b). Three pathways were simultaneously supported by the results from the 10 replicate data sets, with two originating from Germany to Finland and France and a third from New Zealand to Germany (Appendix A). Our phylogeographic analysis shows that Markov rewards for New Zealand (485, 95% credibility interval 207–854) are higher than those for Germany (444, 95% credibility interval 163–808), suggesting that the New Zealand has played an important role in the evolution and persistence of CLRV isolates over the time period studied. In addition, the results from the Markov jump counts showed that out-migration was observed in New Zealand whereas inward bias in France (Figure 2c). This spatial diffusion pattern is concordant with the trade of cherry production, suggesting that movements of CLRV have been associated with human-mediated activities. For example, New Zealand exported more than 1400 tonnes of cherries to Europe, the United States, and Asia in the past decades (http://www.fao.org/faostat/, accessed on 7 September 2022). 

The load and direction of CLRV migrations over time are summarized in Figure 2d. Our results indicated that the first migration event of the virus to Germany from New Zealand were dated to the early 20th century. However, substantial migrations between the two populations were limited to the period of the 1950s to 1990s. The introductions to Finland and France from Germany were inferred to be around 1950s (Figure 2d). In contrast to what was observed for the migrations between countries, internal transmissions within each country, particular in New Zealand, seem to have been more important than those between countries (Figure 2d). Possibly due to transmissions to Finland and France, Germany has experienced a massive decline in native populations since 1980. Taken together, these observations fit well with the migration pathways from our phylogeographic analysis (Figure 2a). Notably, our findings suggest that Germany has been an important hub for the spread of CLRV within Europe. The reason underlying this observation is unclear at present. One possibility is that Germany is a hub for the exchange of cherry or other hosts of CLRV between New Zealand and Europe.

## 4. Conclusions

In summary, this study evaluated the spatio-temporal dynamics of CLRV. Our findings suggest that CLRV originated in New Zealand around the early 20th century and moved to Germany, where it expanded and was further dispersed to other countries of Europe. These data may be potentially valuable for developing efficient and sustainable management strategies for CLRV.

## Figures and Tables

**Figure 1 viruses-14-02179-f001:**
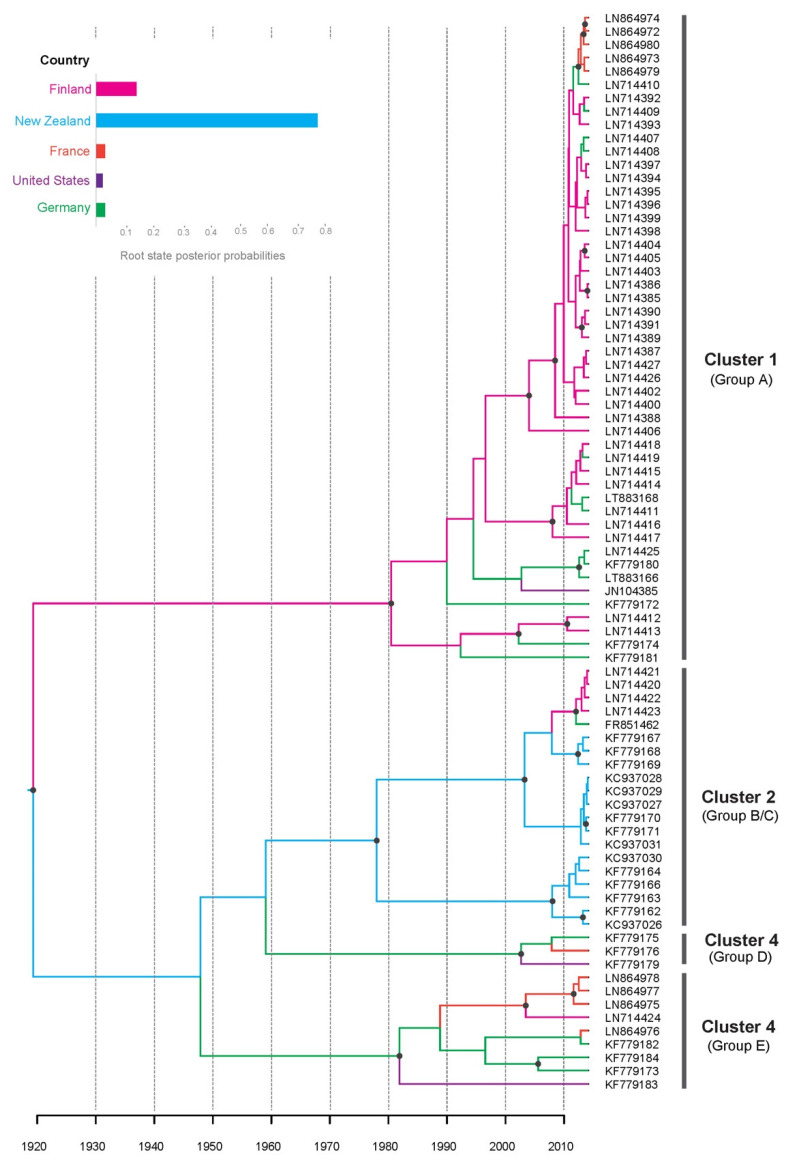
Maximum clade credibility phylogenetic tree summarizing the results of the Bayesian inference. Branch colors indicate inferred region states, as shown in the color key. Only the posterior probabilities (>0.95) are indicated by a black dot on each node. The time axis is scaled to the branch length in units of year. Parentheses below each lineage indicate phylogenetic groups A–E proposed by Rebenstorf et al. (2006).

**Figure 2 viruses-14-02179-f002:**
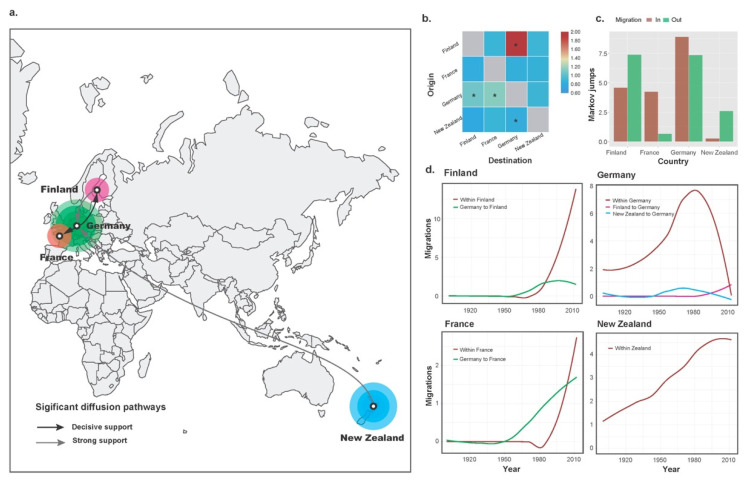
Dispersal patterns of viral migration inferred from cherry leaf roll virus isolates collected from four countries. (**a**) Migration pathways among countries. Arrows represent statistically supported migration rates. (**b**) Heat-maps of the mean migration rate. Significant migration events were marked with asterisks. (**c**) Histogram showing the total number of state counts for migration into and out of each country. The *x*-axis indicates the countries and the *y*-axis indicates the mean Markov jump counts (that is, the number of location state transitions). (**d**) The plots for migration to Finland, Germany, France, and New Zealand over time. The *x*-axes are measured in calendar years and the y-axes indicate migration events on log_10_ scale.

## Data Availability

All data used in this study are publicly available on NCBI. A list of the accession numbers used is found in Appendix A.

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
