# Peer review of "The Spatial Diffusion of Cherry Leaf Roll Virus Revealed by a Bayesian Phylodynamic Analysis"

_viruses, 2022, doi:10.3390/v14102179_

Round 1

Reviewer 1 Report

In present study, the authors investigate the spatial diffusion patterns of cherry leaf roll virus (CLRV) using Bayesian phylodynamics framework based on 81 coat protein gene sequences.  The authors showed that CLRV origined in New Zealand. Interestingly, the authors also found Germany has played an important role in CLRV transmission. Overall, the data of results is solid, the manuscript is well organized and presented. However, several points should be addressed before the MS accepted for publication in Viruses: 

1)        Line 55, “of t CLRV is poorly understood” should be “of CLRV is poorly understood”. “t” should be removed. 

2)        Line 58, “ofToRSV is 2.74×10−3” should be “of ToRSV is 2.74×10-3”. 

3)        MEGA 11 have available downloaded on line. The authors should better used the new version.

Tamura K, Stecher G, and Kumar S (2021). MEGA11: Molecular Evolutionary Genetics Analysis version 11. Molecular Biology and Evolution (https://doi.org/10.1093/molbev/msab120) 

4)        Line 74, “p-value” should be “p-value”. 

5)        Line 181, “(a)Migration pathways” should be “(a). Migration pathways”. 

6)        Line 183, “(c) Histogram” should be “(c). Histogram”. 

7)        Line 185, “(that is, the number of location state transitions) (d). The plots” should be “(that is, the number of location state transitions). (d). The plots”. 

8)        Line 192, “the solid red line.;” should be “the solid red line;”. 

9)        Many editorial errors like above were also found in the MS and references. The authors should rewrite this MS carefully.

Reviewer 2 Report

In this manuscript, Shen et al. investigated the spatial diffusion patterns of CLRV by analysing the coat protein gene sequences collected from five different countries via Bayesian phylodynamics framework. These results revealed CLRV originated in New Zealand about the early 20th century and moved to Germany, where it expanded and became an important hub for the subsequent spread of the virus within Europe. The study is well set out and I have only one point that the authors may consider commenting on, or including in the manuscript:

1. The authors should further discuss the some key results obtained from this study, such as “ Germany became an important hub for the subsequent spread of the virus within Europe.”

Minor comments:

1. Line 58: “2.74×10−3” should be 2.74×10^3.

Reviewer 3 Report

The article by Shen et al. entitled "The spatial diffusion of cherry leaf roll virus revealed by a Bayesian phylodynamic analysis" is an excellent piece of work that describes that how the CLRV, a pathogen originating in New Zealand, migrates to different geographical locations. It is quite interesting to know how a different geographical locale, such as Germany in case of CLRV as mentioned by authors, acts as important source of transmission. It is a good explanation of the fact that how the inter-continental trade can act as carrier of pathogenic microbes. The review is worth accepting in viruses after addressing some minor queries.

There are some typological errors:

Line 58: Please check the spacing error

Line 121: "The" should be replaced with "This"

Line 130: Delete the repeated word "time".

Inclusion of a few lines addressing global economic loss due to CLRV will make it more to the interesting to the readers.

Inclusion of the trend of economic loss in New Zealand (as it is the country of origin) and Germany (as it is playing an important role in transmission) will corelate it well with the economic losses associated with the intercontinental transmission of pathogens. 

The conclusion section is missing. Please incorporate it along with the inclusion of a few future perspectives of the current research.
